# Complete Androgen Insensitivity Syndrome: From Bench to Bed

**DOI:** 10.3390/ijms22031264

**Published:** 2021-01-27

**Authors:** Nina Tyutyusheva, Ilaria Mancini, Giampiero Igli Baroncelli, Sofia D’Elios, Diego Peroni, Maria Cristina Meriggiola, Silvano Bertelloni

**Affiliations:** 1Pediatric and Adolescent Endocrinology, Division of Paediatrics, Azienda Ospedaliero-Universitaria Pisana, 56126 Pisa, Italy; nina.tyutyusheva@gmail.com (N.T.); g.baroncelli@med.unipi.it (G.I.B.); sofia.delios@gmail.com (S.D.); diego.peroni@unipi.it (D.P.); 2Gynecology and Human Reproduction Physiopathology Unit, IRCCS Policlinico di Sant’Orsola, DIMEC, University of Bologna, 40138 Bologna, Italy; mancini.ilaria89@gmail.com (I.M.); cristina.meriggiola@unibo.it (M.C.M.)

**Keywords:** complete androgen insensitivity syndrome, androgen receptor, *AR* gene, gonadal neoplasia, gonadal removal, hormonal substitutive therapy, bone health

## Abstract

Complete androgen insensitivity syndrome (CAIS) is due to complete resistance to the action of androgens, determining a female phenotype in persons with a 46,XY karyotype and functioning testes. CAIS is caused by inactivating mutations in the androgen receptor gene (*AR*). It is organized in eight exons located on the X chromosome. Hundreds of genetic variants in the *AR* gene have been reported in CAIS. They are distributed throughout the gene with a preponderance located in the ligand-binding domain. CAIS mainly presents as primary amenorrhea in an adolescent female or as a bilateral inguinal/labial hernia containing testes in prepubertal children. Some issues regarding the management of females with CAIS remain poorly standardized (such as the follow-up of intact testes, the timing of gonadal removal and optimal hormone replacement therapy). Basic research will lead to the consideration of new issues to improve long-term well-being (such as bone health, immune and metabolic aspects and cardiovascular risk). An expert multidisciplinary approach is mandatory to increase the long-term quality of life of women with CAIS.

## 1. Introduction

Androgen insensitivity syndrome (AIS; OMIM#300068; ORPHA99429; ICD10-E34.5) is a main disorder (or difference) of sex development (DSD) with a 46,XY karyotype [1,2,3]. Albeit rare (with an estimated prevalence of 1:20,000–1:100,000 births) [1,2,3,4], AIS likely represents the most frequent 46,XY DSD, ranging from 40 to 80% in some series [4,5,6,7]. AIS is characterized by the presence of male gonads (testes) in subjects with a female phenotype or with varying degrees of undervirilization of the internal and/or external genitalia [1,2,3].

In this paper, some aspects related to the molecular genetics, diagnosis and management of complete AIS (CAIS) are summarized, and some controversial aspects are discussed, taking into account some findings from basic research.

## 2. Molecular Biology

The androgen receptor (AR) protein belongs to the superfamily of nuclear receptors also designated as NR3C4 (nuclear receptor subfamily 3, group C, member 4). The AR protein consists of a 920 amino acid sequence, has a molecular mass of 110 kDa and is organized in eight exons (indicated as A–H or 1–8) and seven introns. The AR receptor is a single-stranded polypeptide consisting of four main structural domains [1,2,3,8,9]. The N-terminal domain (NTD, 538 amino acids) is encoded by exon 1 and contains the activation function-1 (AF-1) region. It is the transactivating domain, which starts and regulates the transcription of target genes and contributes to the final three-dimensional structure of the receptor [2]. The DNA-binding domain (DBD, amino acids 558–617), encoded by exons 2 and 3, is made up of a high number of cysteine residues which bind two zinc atoms through disulfide bridges, resulting in a tertiary structure called the “zinc finger”, particularly suitable for binding with hormone response elements (HREs). The hinge domain, containing the phosphorylation site for AR, is responsible for androgen-dependent structural changes. The ligand-binding domain (LBD, amino acids 646–920), encoded by exons 4–8, contains specific binding sites for androgens, various transcription factors of coactivation and the activation function-2 (AF-2) region. It promotes the interaction of the receptor with the heat shock proteins (HSPs) in the cytoplasm and then with the androgen hormone, leading to the migration of the AR into the nucleus [2,8,9,10].

A unique feature of the AR receptor is the N-terminal–C-terminal interaction between the AF-1 (N-terminal) and AF-2 (C-terminal) subdomains, aimed to stabilize the connection between the receptor and its ligand and to slow down its dissociation. AF-1 acts in a ligand-independent manner, while AF-2 is ligand-dependent and binds to p160 steroid receptor coactivators such as SRC1, SRC2/TIF2 and SRC3. The homopolymeric traits of amino acids within the NTD (CAG and GGN) are independent modulators of the receptor activity. The three-dimensional structure of the receptor comprises 12 α-helices associated with folded β-sheets; they are arranged as a “tripartite sandwich”. The hydrophobic binding pocket is formed by the helices 3, 4, 5, 7, 11 and 12. Helix 12 (H12) is the outermost α-helix, which folds over the top of the hydrophobic pocket like a box lid. This allows the receptor to “capture” the ligand and hold it, slowing the rate of dissociation, according to an effect called the “mouse trap”. It allows the interaction between the LBD domain, the AF2 subdomain and the LXXLL motif of the associated coregulatory proteins [2,8,9,10].

In baseline conditions, AR resides in the cytoplasm where it forms a multimeric complex with heat shock proteins (HSPs), especially with HSP70, HSP90 and HSP56. After binding with androgens, the receptor dissociates from these proteins, dimerizes and translocates into the nucleus. Nuclear transport is selective and active; it consists of two steps, of which the first does not require energy, while the second step depends on the presence of adenosine triphosphate (ATP). Once in the nucleus, the androgen–AR complex interacts with HREs. The interaction with specific (e.g., ARA24, ARA54, ARA55 and ARA70) and nonspecific (e.g., the SRC and CBP/p300 family of proteins) coactivators and corepressors is fundamental in this complex network [2,3,8,9,10].

According to some *AR* mutation databases, hundreds of pathogenetic mutations related to CAIS are known (Table 1) [11]. Complex rearrangement has been rarely reported.

The majority of *AR* mutations (about two-thirds) are of germline origin inherited from asymptomatic mothers; in other cases, CAIS is due to somatic and de novo mutations [12]. It has been estimated that up to one-third of the de novo mutations may arise in the postzygotic phase, which could partly explain the phenotypic variability observed in subjects with the same genetic defects. In the absence of any mutation in the *AR* gene, but in the presence of the phenotype as well as biochemical data suggesting AIS, an altered signaling pathway due to the impairment of some coactivators or some postligand binding factor has been suggested. However, mutations in cofactor genes have not been found in the large majority of such cases; if so, this hypothesis will require more investigations [13].

## 3. Clinical Features

Females with CAIS present with a normal external female phenotype in girls and women with a 46,XY karyotype and normally functioning testes [1,2,3,8,9]. Psychosexual development is in agreement with female sex [8]. The internal genitalia are absent (“empty pelvis syndrome”) due to the normal action of the anti-Müllerian hormone (AMH) produced by Sertoli cells before birth, which causes regression of the Müllerian structures (uterus, cervix and proximal vagina). Moreover, Wolffian structures do not differentiate because of testosterone resistance [1,2,3]. The testes can be in the abdomen, in the inguinal canal or in the labia majora, causing a bilateral inguinal hernia or labial swelling. These findings are the most frequent clinical signs to suspect CAIS in prepubertal girls. Audi et al. [14] reported inguinal hernia in 47.8% of their sample as a cause for medical advice. Nearly 57% of the CAIS population presented with an inguinal hernia in the U.K. series [15]. In our experience, an inguinal hernia was the cause for referral in more than 30% of cases (17/53). The incidence of inguinal hernias in the pediatric population is 1–4% with a clear prevalence in males (10:1). Thus, karyotypes should be performed in all girls with a mono- or bilateral inguinal hernia [15]. At puberty, there is normal breast development and a typical female distribution of adipose tissue due to androgens being aromatized to estrogens. However, pubic and axillary hair is usually absent or may be scanty. The vagina has a blind bottom, with a length ranging from 2.5 to 8 cm; it is usually adequate for sexual intercourse. Primary amenorrhea, owing to the absence of a uterus, represents the second main reason for medical consultation [1,2,3,8,9].

Today, the diagnosis of CAIS can also arise because of a mismatch between the prenatal sex (based on free-fetal DNA and karyotype analysis) and the phenotype at fetal ultrasound scans or at birth. Some series reported this clinical presentation in about 3% [14], while it is almost doubled in our own experience (5.6%) [4], suggesting an increase of prenatal diagnosis due to larger uses of prenatal screening tests. The need for experienced medical staff to counsel the parents on this delicate area will be urgent. Diagnosis may also be the consequence of a known family history of CAIS (4.3% according to Audi et al. [14] and 13% in our experience [4]).

In consideration of the X-linked transmission, it is recommended to perform a karyotype in prepubertal female relatives of a proband, given the possible recurrence of CAIS within the same family (Figure 1).

A delay between the initial clinical suspicion and the definitive diagnosis is still present despite advances in molecular techniques in our and other’s experience [4,6]. Delayed diagnosis and surgery are performed without seeking a definitive diagnosis in other series [7]. In addition, we found about 10% of misdiagnoses in a group of women with a previous “tag” of CAIS, suggesting that re-evaluation of old diagnoses will enable a better definition of the clinical picture of 46,XY DSD [4].

## 4. Oncological Risk

Malignant transformation of the gonads is the most feared complication in women with CAIS [3,8,9]; timing of gonadectomy to prevent cancer is an issue of debate [16].

Gonadectomy has been performed prior to definitive molecular diagnosis in some females, and early gonadal removal may still occur in prepubertal girls [4,17,18]. However, the oncological risk in children with CAIS is relatively low and remains low until the age of majority (0.02–3%). Deans et al. [19] found that the neoplastic risk is around 0.02% in women under 30 years old and up to 22% in those over that age. Chaurdy et al. [17] reported a neoplastic risk between 0.8 and 22%, with an overall risk of approximately 1.5% in 133 patients over 20 years of age [17]. Thus, gonadal surgery could be likely delayed until complete pubertal development, permitting a spontaneous growth spurt, spontaneous puberty and autonomous decision about surgery after the achievement of the age of majority [18,20]. At any rate, gonadectomy after puberty is still discussed controversially [20].

Postponing gonadectomy until after the age of majority requires an accurate follow-up. Döhnert et al. [20] proposed a regular (bi)annual screening program comprising gonadal imaging by ultrasound or magnetic resonance and the determination of some tumor markers (α-fetoprotein, βHCG, LDH and optionally PLAP in nonsmokers) as well as endocrine evaluation (LH, FSH, testosterone and inhibin B). None of these serological markers are able to detect early neoplastic degeneration of the gonads [21], but the development of specific microRNA assays will be an accurate and sensitive method for the early recognition of a gonadal tumor [21,22]. A major candidate is miR-371a-3p; it is relatively close to being introduced in clinical practice for malignant giant cell tumors (GCTs) [22]. Voorhoeve et al. [23] demonstrated that the members of miR-371-3 can be alternative inhibitors of the p53 pathway. MiR-371a-3p is highly informative in identifying the malignant component of all GCTs except teratoma compared to the standard α-fetoprotein and βHCG. It can be detected in serum, plasma and cerebrospinal fluids [23]. MiR-375 has been suggested to be diagnostic for teratoma as well, although it is not clinically proven so far [24].

## 5. Hormonal Substitutive Therapy

Hypergonadotropic hypogonadism is obviously present in adolescent and adult females with CAIS and removed gonads [1,2,3]. Adequate hormonal replacement therapy (HRT) is mandatory for these patients. In adolescents who underwent surgery before puberty, HRT should assure the development of secondary sexual characteristics, i.e., breasts; a normal pubertal growth spurt and body proportions; adequate muscle and fat mass development; optimal bone mineral accumulation; and psychosocial and psychosexual maturation, as well as satisfying general well-being [25,26]. HRT is necessary in women undergoing surgery during or after adolescence to complete or maintain female secondary sexual features, to prevent bone loss and neurocognitive disorders and to guarantee cardiovascular health. Thus, HRT should be assured at least until the average of natural menopause in 46,XX women [25,26].

There is no unique HRT protocol or any evidence-based data on the optimal hormone formulations, route of administration, doses and monitoring parameters in women with CAIS. Various formulations of estrogens are used to induce puberty or adult HRT. The more adequate formulations are likely oral micronized or transdermal (gel or patch) bioidentical 17-β-estradiol [25,26]. The latter permits a better personalization of estrogen dose and guarantees a more physiologic delivery with controlled absorption, more constant plasma levels, improved bioavailability, reduced side effects due to decreased first-pass effects of liver metabolism, less interference with IGF-1 levels and a painless and simple mode of administration. However, the optimal dose of HRT in young women with CAIS remains unknown. The dosage of estrogens usually used for substitutive therapy in 46,XX hypogonadic women may be simply not applicable to adolescents or young adults with CAIS [26]. The adherence to HRT is sometimes not optimal. Lack of adequate information and/or understanding of the indications, mechanism of action and side effects of HRT could play a role [27]. The symptoms of not undergoing HRT are not immediately obvious, jeopardizing long-term compliance in not well-motivated women [26].

Because testosterone is the main steroid hormone secreted by testes in women with CAIS, it may be an alternative therapy to estrogens for these females [25,26]. Recently, a multicenter, randomized double-blind trial explored transdermal estrogens (1.5 mg/day) vs. transdermal testosterone (50 mg/day), showing that the latter was well tolerated and as safe as the former [28]. No virilization occurred and gonadotrophin concentrations remained stable in both groups. More adverse events were found in patients receiving testosterone when compared to the estrogen treatment group, but limited serious adverse effects were reported in both groups [28]. Further data on larger samples, longer follow-up and evaluation of somatic parameters (e.g., bone mineralization, body composition and metabolic profile) are needed to evaluate the benefits of testosterone therapy in CAIS.

Progestins are not indicated due to the absence of a uterus and possible negative effects on bone mineral density (BMD) [26], but several women have still undergone combined estrogen plus progestin therapy, at least in Italy [4].

International surveys should be performed to give evidence-based indications for optimal HRT in women with CAIS and removed gonads. Specific hormonal profiles of these women should be considered when monitoring HRT [29]. Since the decline in testicular function is smoothed in comparison with ovarian menopause, the optimal long-term HRT with aging should be explored, too.

## 6. Bone Health

Androgens are involved in the growth, development, peak achievement and maintenance of bone mass [30]. Thus, androgen resistance may impair bone health [30].

Experimental data in AR knockout (ARKO) mice demonstrated a phenotype with reduced bone mass and decreased stiffness. Microstructural analysis revealed a significant loss of bone volume and trabecular number with a shift towards bone resorption [31]. In 1989, Colvard et al. [32] discovered *AR* expression in cultured osteoblasts, which was later confirmed both in vivo and in vitro in osteoblasts and osteocytes. Other authors underlined the role of androgen-enhanced tissue-nonspecific alkaline phosphatase expression mediated by *AR* for the initial formation of hydroxyapatite crystals during osteoblastic mineralization. AR was found to preserve the number of trabeculae independently from GH/IGF1 effects, acting directly on osteocytes or indirectly by inhibiting osteoclastogenesis through interaction with osteoblast precursors, but it seems to have no direct role on osteoclasts [30,31]. Gonadectomy represents a major risk factor for bone health due to iatrogenic hypogonadism (see above). Other factors possibly affecting bone health in CAIS are summarized in Table 2.

Few data are available on bone health in CAIS (Table 3), due to the rarity of this condition, which makes the analysis of large and homogeneous series difficult.

Some studies have shown reduced BMD in women with CAIS according to both female and male reference values (Table 3). Study samples are usually small and heterogeneous in terms of gonadal status and type of HRT. BMD is measured with different methodologies often not comparable to each other. In some studies, the diagnosis is based on clinical data and on the karyotype without molecular analysis of *AR* gene (Table 3) with the risk of including patients with other forms of 46,XY DSD [4]

The effectiveness of HRT in normalizing BMD values in gonadectomized women with CAIS remains an unsolved issue [36]. Recently, Gava et al. [45] demonstrated a small but significant increase in lumbar BMD, while femoral and total body BMD did not significantly change after 4–6 years of HRT in a cohort of gonadectomized women with CAIS (Figure 2). Total body BMD was higher in the group that used 2 mg of transdermal estradiol gel than in the group receiving oral formulation (estradiol valerate 2 mg) [38]. This finding could be explained by impaired liver IGF-1 synthesis with the lack of its trophic effects on bone tissue in the group treated with the oral formulation. However, BMD did not reach values of age-matched 46,XX healthy control women [38] (Figure 2). Estradiol dose may be involved. Gonadectomized women with CAIS may receive relatively low estrogen for females who should be in full activity of their reproductive axis, and AR resistance may be an additional factor [26]. Thus, a more patient-centered treatment should be taken into account. Taes et al. [46] demonstrated an improvement in BMD in one gonadectomized woman with CAIS treated with high-dose estrogen (from 0.3 to 2.25 mg/day in the first year, then 3 mg/day for 4 years), although lumbar BMD still remained at −2 SD at the end of the 5 year follow-up period. Higher-than-usual substitutive doses of estrogens should be considered in women with complete androgen resistance to optimize bone health [26,46], assuring that they undergo strict clinical and laboratory follow-up.

## 7. HRT and Psychological Well-Being

Despite the administration and good adherence to classic HRT, many women with CAIS reported a decrease in psychological well-being and sexual functioning after gonadectomy ([47] and personal unpublished data). Central regulation of sexual functioning is complex and the role of testosterone, if any, in improving sexual desire in women with CAIS is largely unclear. Testosterone is likely converted by aromatase or 5-α reductase in the brain into estradiol and 3 α-androstanediol, respectively [48]. These hormones may be involved in activating sexual behavior and preserving the central effects of testosterone in these women. Thus, testosterone may represent an alternative to estrogens for HRT after surgery [25,26,28]. One trial with four patients did not demonstrate that androgen replacement therapy is preferable to usual estrogen treatment with regard to psychosexual functioning [49]. The recent German double-blind and randomized crossover trial showed no significant differences in terms of psychological well-being, mental health and quality of life between therapy with estrogens or testosterone, but testosterone was superior to estrogens in improving sexual desire [28].

Further longitudinal randomized trials are needed to identify the advantages of testosterone replacement therapy with respect to classical estrogen therapy in terms of psychological and sexual well-being in women with CAIS.

## 8. Immune and Metabolic Aspects

Recent studies of ARKO mouse models uncovered new cell-type- or tissue-specific actions of AR [50].

A decrease in myelocytes/metamyelocytes and mature neutrophils were found in bone marrow cells of ARKO mice. The defect in granulopoiesis occurs during the transition between the proliferation of precursors and the maturation of neutrophils; such defects together with a high rate of apoptosis lead to increased susceptibility to acute bacterial infection [51]. Impaired chemokine receptor CXCR-2 mediated neutrophil migration but not degranulation, as observed in ARKO mice models [50]. A reduced number of monocytes and macrophages in ARKO mice indicates that AR is involved in the modulation of inflammation, including inflammatory-associated atherosclerosis. ARKO mice were found to contain higher levels of serum IgG2, IgG3 and basal anti-double-stranded DNA IgG antibodies with an increased risk of collagen-induced arthritis and other autoimmune diseases. AR is also expressed in thymic epithelial reticular cells, thymocytes and other stromal cells, and it is an important modulator of thymocyte development [50]. Data on these issues are lacking in women with CAIS, and they should be addressed.

Impaired AR signaling can be responsible for an increased risk of metabolic syndrome, diabetes and cardiovascular diseases. ARKO mice became obese with increased body white adipose tissue mass without an increase in food uptake or dyslipidemia at ages beyond 10 weeks old; hyperinsulinemia, hyperleptinemia, hyperglycemia, hypoadiponectinemia, increased serum levels of triglycerides and free fatty acids and impaired glucose tolerance occurred with aging [52]. These data suggest that AR loss may induce insulin and leptin resistance and dysregulation of lipid metabolism, favor adipocyte differentiation and fat deposition, leading to obesity. Moreover, ablation of neuronal AR resulted in hypothalamic insulin resistance, which leads to systemic insulin resistance, dysregulation of glucose homeostasis and lipid metabolism and visceral obesity. The ablation of AR in the pituitary gland resulted in dysregulation of feedback control of glucocorticoid production, which also led to obesity in a mouse model [50].

The ARKO mice model presents with a decreased heart size, smaller volume and wall thickness of the left ventricle and significantly reduced aortic NO synthase expression. Taken together, these observations seem to indicate that AR disruption may play some role in promoting atherosclerosis. It also seems that the AR might play some role in the initiation or progression of abdominal aorta aneurism [50].

Data on these items are largely absent for humans with CAIS. Some of our previous data in a small group of women with CAIS and intact or removed gonads suggested that altered AR signaling may increase body fat and affect some metabolic parameters [53], but these findings were not confirmed in a subsequent series of adult patients with removed gonads [45]. One young woman with a mild overweight status in our series suddenly died of acute vascular disease at the age of 38 years. Thus, assessment of body composition, metabolic profile and cardiovascular risk should be highlighted in women with CAIS during aging and related to their hormonal status.

## 9. Conclusions

AR plays key roles during male sex differentiation in utero and produces pleiotropic effects during extrauterine life; the understanding of these roles and effects has been expanded by recent basic researches [1,2,3,50]. In humans, the new acquisitions must be considered to optimize the health (“a state of complete physical, mental and social well-being and not merely the absence of disease or infirmity” as per WHO, 1948) of women with CAIS. To do this, a multidisciplinary and multicenter approach is required to overcome the rarity of this DSD, and this should include both physical and mental healthcare [54]. Documented experts in pediatric endocrinology and gynecology are central specialists within the multidisciplinary teams.

## Figures and Tables

**Figure 1 ijms-22-01264-f001:**
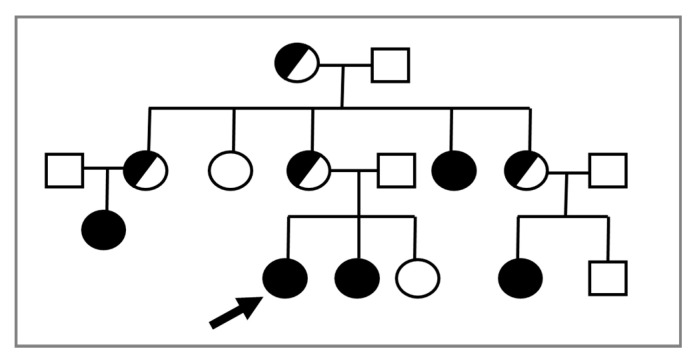
The proband (indicated by the arrow) is a young 46,XY woman (25 years old) with CAIS, in whom the *AR* variant p.Trp797* was found. Subsequently, the same mutation was detected in the 46,XY sister with CAIS (⬤) and in other affected 46,XY family members with the same DSD (⬤). Female 46,XX carriers were suspected but not proven by molecular analysis of *AR*.

**Figure 2 ijms-22-01264-f002:**
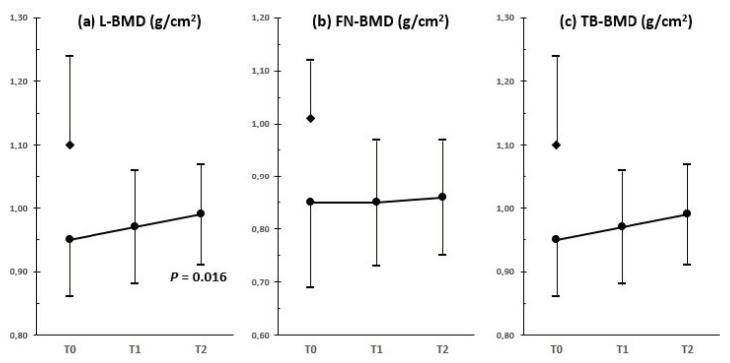
Bone mineral density (BMD) assessed by DXA at (**a**) lumbar spine (L), (**b**) femoral neck (FN) and (**c**) total body (TB) in a sample of 18 women with CAIS (⬤) during HRT (17-β-estradiol 2 mg/daily) (T0 = baseline data, T1 = 1–3 years, T2 = 4–6 years). Control group (◆) consisted of 32 healthy age-matched 46,XX women (drawn from 45).

**Table 1 ijms-22-01264-t001:** *AR* genetic variants in complete androgen insensitivity syndrome (CAIS): personal experience in comparison with international databases [11].

Type of Genetic Variants	Pisa ^a^	McGill ^b^	HGMD ^c^
Point missense or nonsense mutations	65.1%	50.0%	75.0%
Insertions or deletions	12.1%	28.0%	6.6%
Intronic or intron–exon junction point mutations	12.15%	5.4%	4.0%
Small or gross deletions of the *AR* gene	7.6%	6.7%	14.4%
Complete deletions of the *AR* gene	3.0%	1.3%	¾

Genetic variants associated only with CAIS (no partial or minimal AIS considered): ^a^
*n* = 66; personal series partly published in 4; ^b^ McGill University *AR* database: *n* = 314, (www.mcgill.ca/androgendb); ^c^ HGMD (Human Gene Mutation Database): *n* = 533 (www.hmgd.cf.ac.uk).

**Table 2 ijms-22-01264-t002:** Some mechanisms involved in impaired bone health in females with CAIS [30,31,32,33,34,35,36].

Factor(s)	Mechanism(s)
Complete AR resistance	Abnormal activity of osteoblasts and osteocytes.
Iatrogenic hypergonadotropic hypogonadism	Inadequate hormonal substitutive therapy.
	Absence of testicular INSL-3 (reduced bone mass through dysregulation of osteoblastic differentiation, deposition of the bone matrix and osteoclastogenesis; altered functioning of the musculo-skeletal unit in postnatal life).
	High FSH levels (greater bone resorption by osteoclasts due to the activation of the FSH receptor *)
Others	Age at gonadectomy (before or after achievement of peak bone mass).
	Type, doses, compliance and duration of HRT.
	Unhealthy eating, low calcium intake, reduced physical activity, inadequate muscle mass, smoking and low vitamin D.
	Aging.

* Anti-FSH receptor monoclonal antibodies in ovariectomized mice are able to improve bone health through downregulation of FSH signaling [37].

**Table 3 ijms-22-01264-t003:** Main data on bone mineral density (BMD) status in women with CAIS (single case reports are not shown).

Author	*n*	Age	Genetic Analysis	HRT Compliance	BMD		
					Lumbar	Femoral	R.V.
Soule et al. [38]	6	13–38	no	v.c. ^a^	↓↓	↓↓	F
Bertelloni et al. [39]	10	4–20	yes	b.g./v.c.	↓↓ ^b^	ND	F/M
Marcus et al. [40]	22	11–65	no	v.c.	↓ ^d^	N ^d^	F
Tian et al. [41]	14				↓↓	↓↓	F/M
Sobel et al. [42]	12	17–62	yes/no	b.g./v.c.	↓↓ ^c^	↓ ^c^	F/M
Danilovic et al. [43]	5	20–25	yes	good/fair	↓ ^b^	N ^b^	F/M
Han et al. [44]	46	32.2	yes/no	good	↓↓	↓	F
Gava et al. [45]	32	19–57	yes	good	↓	↓	F

r.v. = reference values used to evaluate BMD, v.c. = variable compliance with HRT, b.g. = before gonadectomy, F = female sex, M = male sex, ↓↓ = very low, ↓ = low, ND = not done, N = normal. ^a^ Variable degrees of sex-steroid deficiency in the years before BMD measurement. ^b^ Corrected for apparent bone volume by mathematical formulas. ^c^ Self-reported diagnosis (no biochemical or genetic data available to support diagnosis). ^d^ Corrected for body weight, reduced after correction for apparent volumetric BMD.

## Data Availability

This review did not report any original data.

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
