# Peer review of "Complete Androgen Insensitivity Syndrome: From Bench to Bed"

_ijms, 2021, doi:10.3390/ijms22031264_

Round 1

Reviewer 1 Report

This is very interesting, well written,  review that comprises all necessary information regarding complete androgen insensitivity. 

I don't have any special comments but one: to correct the sentence in the abstract (row 16) starting from: "The molecular basis..."

Author Response

The authors thank the reviewer for thr positive comments and for the constructive criticisms.

Reply:

  1. linguage and spelling has been revised, as required.
  2. Abstract, line 16: the sentence “The molecular basis …” has been rewritten (see text).

Reviewer 2 Report

The manuscript by Tyutyusheva et al descibres CAIS from various aspects. It is well structured and written, the critisim does not question the merits of the work.

Major issues.

  1. The mentioned locus-specific database has not been updated recently according to the authors. Therefore, a thorough analysis of the different othar available databases is neccessary (ClinVar, HGMD Professional). Data from them should be added to Table 1.
  2. A cartoon of the AR that includes the domain structure and also the missense mutations would help the reader to follow the text.

Minor issues.

  1. Consent to publish the photo of the patient in Fig 1 should be mentioned.
  2. A noncense mutation should be designated either by '*' or 'Ter' rather than 'X' according to the recent HGVS recommendations (Fig 2).

Author Response

The authors thank the reviewer for the positive comments and for the constructive criticisms.

  1. English language and spelling has been revised, as required.
  2. Table 1: data on genetic variations in CAIS from HGMD have been added.
  3. We thank for this suggestion, anyway a comprehensive cartoon of the AR mutations can be easily obtained from literature or online database according to references cited in the text. So, we think to be unnecessary for the present paper.
  4. We thank again for this suggestion. The sentence “(written informed consent from parents was obtained to publish the photos)” has been added in the legend of figure 1.
  5. Figure 2: the designation of AR variant p.Trp797* is now rightly written.